# Review: Tissue Engineering of Small-Diameter Vascular Grafts and Their In Vivo Evaluation in Large Animals and Humans

**DOI:** 10.3390/cells10030713

**Published:** 2021-03-23

**Authors:** Shu Fang, Ditte Gry Ellman, Ditte Caroline Andersen

**Affiliations:** 1Laboratory of Molecular and Cellular Cardiology, Department of Clinical Biochemistry and Pharmacology, Odense University Hospital, J. B. Winsløwsvej 25, 5000 Odense C, Denmark; dellman@health.sdu.dk (D.G.E.); DAndersen@health.sdu.dk (D.C.A.); 2The Danish Regenerative Center, Odense University Hospital, J. B. Winsløwsvej 4, 5000 Odense C, Denmark; 3Institute of Clinical Research, University of Southern Denmark, J. B. Winsløwsvej 19, 5000 Odense C, Denmark

**Keywords:** small-diameter tissue engineered vascular grafts (SD-TEVGs), large-animal models, patency, end-to-side anastomosis, end-to-end anastomosis, antithrombotic therapy

## Abstract

To date, a wide range of materials, from synthetic to natural or a mixture of these, has been explored, modified, and examined as small-diameter tissue-engineered vascular grafts (SD-TEVGs) for tissue regeneration either in vitro or in vivo. However, very limited success has been achieved due to mechanical failure, thrombogenicity or intimal hyperplasia, and improvements of the SD-TEVG design are thus required. Here, in vivo studies investigating novel and relative long (10 times of the inner diameter) SD-TEVGs in large animal models and humans are identified and discussed, with emphasis on graft outcome based on model- and graft-related conditions. Only a few types of synthetic polymer-based SD-TEVGs have been evaluated in large-animal models and reflect limited success. However, some polymers, such as polycaprolactone (PCL), show favorable biocompatibility and potential to be further modified and improved in the form of hybrid grafts. Natural polymer- and cell-secreted extracellular matrix (ECM)-based SD-TEVGs tested in large animals still fail due to a weak strength or thrombogenicity. Similarly, native ECM-based SD-TEVGs and in-vitro-developed hybrid SD-TEVGs that contain xenogeneic molecules or matrix seem related to a harmful graft outcome. In contrast, allogeneic native ECM-based SD-TEVGs, in-vitro-developed hybrid SD-TEVGs with allogeneic banked human cells or isolated autologous stem cells, and in-body tissue architecture (IBTA)-based SD-TEVGs seem to be promising for the future, since they are suitable in dimension, mechanical strength, biocompatibility, and availability.

## 1. Introduction

The leading cause of death worldwide is cardiovascular disease [1]. In the European Union countries, 119 deaths per 100,000 inhabitants in 2016 were caused by ischemic heart diseases [2]. The latter is most often caused by atherosclerosis, which also results in peripheral artery disease. The involved artery is narrowed in lumen, and the flow rate is limited, resulting in reduced blood perfusion, and oxygen and nutrients supply. Due to the development of improved medication and percutaneous intervention, surgical intervention has decreased in some areas of the world; however, bypass grafting still plays an important role for severely affected patients to recover blood perfusion.

For coronary-artery bypass grafting (CABG), the most optimal graft is autologous left internal mammary artery [3], which offers adequate diameter and length for coronary-artery revascularization [4], with a satisfying long-term patency rate of more than 85% after 10 years [5] (Table 1).

The main failure reason, in the late phase, for left internal mammary artery graft is competitive flow from residual blood flow from the native coronary artery [6]. In contrast, the suboptimal, but most commonly used graft, is saphenous vein that displays a relatively low long-term patency rate of 61% after 10 years [6]. It often fails due to thrombosis in the early phase (within 1 month), whereas intimal hyperplasia and atherosclerosis are the failure reasons in intermediate (within 12 months) and late phases (after 12 months) [7]. Other autologous arteries (e.g., radial artery and right gastroepiploic artery) may be used alternatively for CABG; however, no prosthetic graft is approved for CABG yet [4].

For bypass grafting in lower extremity, infrainguinal bypass above the knee (femoropopliteal bypass) is considered to be a medium-diameter surgery, while infrainguinal bypass below the knee (femorodistal bypass) is considered to be a small-diameter bypass surgery (Table 1). Although the autologous saphenous vein displays a diameter usually smaller than 6 mm, it still remains the most optimal graft for both above- and below-knee bypass surgery due to the unavailability of autologous arterial graft in general [8], but it should be noted that the primary patency rate is 53.7% after 3 years [9]. Mechanisms of saphenous vein graft failure in infrainguinal bypass are suggested to be similar to those in CABG [10]. However, unlike CABG, other non-autologous grafts (e.g., prosthetic grafts and human umbilical veins) are available for lower extremity bypass grafting above the knee with relative lower, but still comparable, primary patency rates [8]. Small-diameter bypass grafting is also performed in upper extremity but with much less incidence than bypass grafting in the heart and the lower extremities [11].

In general, arterial bypass grafting in the heart or below the knee requires small-diameter grafts. Thus, shortage of material for such surgeries remains a big challenge because autologous grafts are often not available in certain patient groups such as claudicants, patients with diabetics or vein disease, and in patients requiring reoperations. This has further underscored the need for developing alternative small-diameter vascular grafts [12,13]. One candidate, small-diameter tissue-engineered vascular grafts (SD-TEVGs), is fabricated using novel techniques and interdisciplinary knowledge including material, engineering, and cell biology. Advantages of using SD-TEVGs as compared to autografts, include noninvasive surgery during preparation of grafts, unlimited availability, and customized dimension.

**Table 1 cells-10-00713-t001:** Medium- and small-diameter arterial bypass grafting in clinical practice.

Diseases	Bypass Site	Host Artery Diameter (mm)	Optimal Graft	Graft Length (cm)	Graft Diameter (mm)	Anastomotic Configuration (Distal)	1-Year Patency	3-Year Patency	10-Year Patency
**Coronary-artery disease (CAD)**	Coronary-artery bypass	P: 1.6–7.2 M: 1.0–6.7 D: 0.8–2.5 * [4]	Left internal mammary artery [3]	14.3–19.5 [4]	1.5–1.8 [4]	End-to-side	95% [5]	93% [5]	85% [5]
**Peripheral arterial disease** **(PAD)**	Infrainguinal bypass	Femoral: P: 10.2 D: 7.7 Popliteal: 6.9 Tibial: 3.8/4.2 # [14]	Great saphenous vein [15]	72.4 ± 6.6 [16]	P: 5.2 ± 0.6 M: 3. 3 ± 0.5 D: 1.7 ± 0.3 [16]	End-to-side	74.4% [9]	53.7% [9]	

* P: proximal segment; M: media segment; D: distal segment; and # Tibial: anterior/posterior.

## 2. SD-TEVGs Evaluated in Humans

In past decades, different types of SD-TEVGs have indeed been explored and evaluated in humans, either as arterial bypass grafts or arteriovenous shunts. To exemplify current progress, representative SD-TEVGs tested in humans are summarized in Table 2 and below.

There are several case reports and clinical trials that investigated the usage of synthetic SD-TEVGs at the aortocoronary site.

In 1976, Sauvage et al. reported about a knitted Dacron filamentous vascular prosthesis (3.5 mm in diameter and 4 cm long) as an interposition graft at the aortocoronary site in a 65-year-old patient to repair the coronary artery after removal of a saccular aneurysm in the ascending aorta [17]. This graft maintained patency during the 16-month follow-up period. Success has also been observed in similar synthetic grafts at the aortocoronary site [18]. However, considering the bypass location between the aorta and the proximal end of the coronary artery with high flow, these case reports did not support implantation of synthetic graft for common CABG, where the grafts need to be implanted to coronary arteries at more distal positions [17].

From 1982 to 2008, at least six types of grafts were further evaluated in patients that underwent CABG [19] (Table 2):

(1) glutaraldehyde-fixed human umbilical vein grafts with a patency of 46% after 3–13-month follow-up published in 1982 [20];

(2) cryopreserved allograft saphenous vein with a patency of 41% after 2–16-month follow-up published in 1992 [21];

(3) dialdehyde starch-treated bovine internal thoracic artery grafts with a patency of 16% after 3–23-month follow-up in 1993 [22];

(4) No-React bovine internal mammary artery with a patency of 57% after 1–4.5-year follow-up in a study in 2004 [23] and a patency of 23% after 3–11-month follow-up in another study in 2008 [24];

(5) autologous endothelial cell-seeded expanded polytetrafluoroethylene (ePTFE) grafts with a patency of 90.5% after 7.5–48-month follow-up in 2000 [25];

(6) de-endothelialized and cryopreserved allograft veins seeded by autologous endothelial cells with a patency of 50% after 9-month follow-up published in 2001 [26] and 0% patency after 32 months, published in 2019 [27].

The first four types of grafts showed very poor patency and therefore were not recommended as alternative choices for CABG in patients, whereas the fifth type of graft displayed high patency [25], suggesting promising improvement of graft patency by endothelialization as also discussed below. This improvement of endothelialization was also seen in the allograft veins seeded by autologous endothelial cell [26,27], as compared to the similar cryopreserved allograft saphenous vein but without endothelialization [21]. However, when comparing the two types of grafts that were both endothelialized [25,26,27], the synthetic ePTFE [25] seems much better than the cryopreserved allograft [26,27], indicating that elimination of immunogenicity in the allografts cannot be fully achieved by cryopreservation and therefore need to be further improved by using other methodology such as decellularization. Thus, until now, modified synthetic, allogeneic, or xenogeneic grafts have indeed been studied in humans for CABG, however with very limited success due to their thrombogenicity. Furthermore, there are no human studies testing SD-TEVGs for CABG initiated after 2008.

In regard to human studies for artery bypass grafting below the knee, Almasri et al. reviewed large-scale clinical trials in 2018 and revealed a primary patency around 50% of FDA-approved prosthetic grafts (cryopreserved saphenous vein allografts and heparin bounded polytetrafluoroethylene (PTFE)) at 1-year follow-up using meta-analysis [28]. Recently, another type of FDA-approved TEVG termed crosslinked bovine carotid artery graft (BCAG) has been examined in patients for artery bypass grafting below the knee. They display a long-term primary patency at 50–75% 5 years after implantation (Table 2) [29], which is comparable to autologous vein graft and might be better than synthetic grafts [28]. However, the study was retrospective, and therefore, prospective randomized studies are needed to compare these xenogeneic grafts with autologous vein grafts and synthetic grafts. To reduce the thrombogenicity of synthetic grafts, Williams et al. recellularized ePTFE with autologous adipose-derived stromal vascular fraction cells and implanted these modified grafts as femoral-to-tibial bypass grafts in a phase 1 clinical trial (Table 2) [30]. The 1 year patency of these recellularized grafts was 60% (3/5 were patent).

**Table 2 cells-10-00713-t002:** Small-diameter tissue-engineered vascular grafts (SD-TEVGs) evaluated in humans.

Author	Graft Type	Year	Graft	Number of Patients	Recellularization	Follow-Up Time	Primary Patency
**CABG**							
Silver [20]	Allogeneic	1982	Glutaraldehyde-fixed human umbilical vein grafts	11	None	3 to 13 months	46%
Laub [21]	Allogeneic	1992	Cryopreserved allograft saphenous vein	19	None	2 to 16 months	41%
Mitchell [22]	Xenogeneic	1993	Dialdehyde starch-treated bovine internal mammary artery	18	None	3 to 23 months	16%
Reddy [23]	Xenogeneic	2004	No-React bovine internal mammary artery	7	None	1 to 4.5 years	57%
Englberger [24]	Xenogeneic	2008	No-React bovine internal mammary artery	17	None	3 to 11 months	23%
Laube [25]	Autologous cells on synthetic	2000	Autologous endothelial cell-seeded ePTFE graft	14	Autologous endothelial cell	7.5 to 48 months	91%
Lamm [26] and Herrmann [27]	Autologous cells on allograft	2001 and 2019	Deendothelialized/cryopreserved allograft veins seeded by autologous endothelial cells	12	Autologous endothelial cell	16 to 18 years	80% (6 months); 50% (9 months); 0% (32 months)
**Bypass grafting below knee**							
Lindsey [29]	Xenogeneic	2017	Crosslinked bovine carotid artery	80	None	5 years	52% to 75%
Williams [30]	Autologous cells on synthetic	2017	Adipose-Derived Stromal Vascular Fraction Cell seeded ePTFE	5	Adipose-Derived Stromal Vascular Fraction Cell	1 year	60%
**AV shunt for hemodialysis access**							
Kennealey [31]	Xenogeneic	2011	Crosslinked bovine carotid artery	26	None	1 year	61%
Harlander-Locke [32]	Xenogeneic	2014	Crosslinked bovine carotid artery	17	None	18 months	73%
Wystrychowski [33]	Allogeneic	2014	Allogeneic cell sheet-based TEVG, dehydrated	3	None	<11 months	9.5 patient-month of use
Lawson [34]	Allogeneic	2016	Allogeneic human acellular vessels	60	None	>1 year	28% at 12 months
L’Heureux [35]	Autologous	2007	Autologous cell sheet-based TEVG	6	Autologous fibroblast and endothelial cells	<13 months	24 patient-months of use
McAllister [36]	Autologous	2009	Autologous cell sheet-based TEVG	10	Autologous fibroblast and endothelial cells	>6 months	68 patient-months of use
Wystrychowski [37]	Autologous	2011	Autologous cell sheet-based TEVG, cold-preserved	1	Autologous endothelial cells	8 weeks	8 patient-weeks of use
SD-TEVGs: Small-diameter tissue engineered vascular grafts; CABG: coronary-artery bypass grafting; AV shunt: arteriovenous shunt. There might be other similar studies not included here.							

Instead of CABG and bypass grafting in lower extremity, arteriovenous shunt for hemodialysis in patients with end-stage renal disease has become a popular model for testing novel TEVGs (Table 2), since adverse events like graft failure are less likely to harm these patients. Since 2007, L’Heureux and colleagues have focused on testing novel cell-sheet-based TEVGs in patients as hemodialysis access. As such, they have assessed autologous fully recellularized TEVGs [35,36], autologous cold-preserved TEVGs with endothelial cells [37], and allogeneic dehydrated TEVGs [33]. However, in all three types of grafts either poor mechanical properties or poor patency outcome were apparent, as dilation, aneurysm, and thrombus were often observed. In contrast, the TEVGs developed by Lawson and colleagues, using cell-sheet-based technology and decellularization, manifested stable mechanical strength over time. Although a poor primary patency was observed in this study, the secondary patency of the cell-sheet-based TEVGs was found fairly positive at 89% after more than 1 year follow-up. Moreover, as compared to the synthetic grafts tested, the cell-sheet-based TEVGs possessed higher resistance to prosthetic infection, which is a common reason for graft failure in arteriovenous shunt for hemodialysis access [34]. Crosslinked BCAG has also been suggested as an alternative to autologous grafts. When implanted in patients as arteriovenous shunt for hemodialysis access, crosslinked BCAG exhibit a patency of 60% to 70% after 12 or 18 months [31,32], which is similar to the positive outcome observed in lower extremity bypass grafting [29].

Thus, although some progress has been achieved regarding SD-TEVGs in clinical studies, autologous arteries or veins are still superior and the first choice for small-diameter artery bypass grafting. However, techniques in this field develop at a high speed (see below), and progress is substantiated by the large number of studies testing SD-TEVGs in large animals.

## 3. SD-TEVG Studies in Large-Animal Models

### 3.1. Systematic Search

Large-animal models are regarded as important preclinical tools to determine the quality and functionality of novel SD-TEVGs. They are more similar to humans with regards to the coagulation system, hemodynamics, and hematological profiles as compared to small animals [38,39], although rodent and rabbit models are widely used to test SD-TEVGs as an alternate to human and large-animal models [40,41]. To further provide a status overview on the field of SD-TEVGs, we therefore used a systematic approach to identify studies that investigate SD-TEVGs as a relative long piece (longer than 10 times the inner diameter) in large-animal models. Accordingly, studies from 1 January 1995, until 1 November 2019, were found on PubMed by searching: “vascular graft”, “tissue engineering”, “smameter”, and “in vivo” according to inclusion and exclusion criteria that apply to a strategy of clinical translation (Table 3).

Specifically, earlier studies have suggested that a limited length (<10 times the inner diameter) might overevaluate the patency of grafts [42,43]. In addition, considering the clinical need for substitutes of relative long (>20 cm) TEVGs for bypass grafting, we only included grafts with a length more than 10 times the diameter, which are believed to be much more clinically relevant regarding to bypass grafting in humans [42]. Furthermore, considering the high flow rate at the aorta site and low pressure in the venous and pulmonary system, only studies concerning arterial small-diameter engraftment sites like the coronary-artery bed and medium-diameter sites like the carotid or femoral vasculature were included (Table 3). In this regard, it is important to note that SD-TEVGs are required for femerodistal or femeropopliteal bypasses, even though the femoral artery possesses a diameter larger than 7 mm (Table 1).

A total of 39 studies were included (Table 4, Table 5, Table 6 and Table 7), and information regarding model-related conditions (animal, bypass site, anastomosis fashion, and usage of antithrombotic treatment), graft-related parameters (material, origin, modification, recellularization, and mechanical modification) and outcome (follow-up, patency, and failure mechanism) were extracted and stratified (Table 4, Table 5, Table 6 and Table 7). Notably, some studies represented more than one type of TEVG or more than one insertion site, etc., and data are thus categorized separately in our stratifications.

Regarding insertion site, 32 studies used arterial bypass (at coronary-artery sites (1 study), in carotid (27 studies), as femoral arteries (six studies) (Table 4, Table 5 and Table 6)), whereas nine studies concerned arteriovenous shunts (Table 7). Furthermore, in studies using arterial bypass, six studies used an end-to-side (ETS) anastomotic surgery technique while 25 performed end-to-end (ETE), and one study did not specify regarding this (Table 4, Table 5 and Table 6). In contrast, ETS was most often performed in arteriovenous shunt due to its anatomical nature (Table 7). Overall, 26 studies used systemic antithrombotic treatment, while nine studies did not use such treatment, and two studies included evaluation of grafts under both conditions for direct comparisons [44,45], while two studies did not reveal if systemic antithrombotic treatment was applied [46,47] (Table 4, Table 5, Table 6 and Table 7). Using these stratifications (Table 4, Table 5, Table 6 and Table 7), we discuss below the different types of SD-TEVGs and their potential for future clinical translation.

### 3.2. Tissue Engineering of Small-Diameter Vascular Grafts

In general, tissue engineering is referred to as an interdisciplinary field that combines biological and engineering knowledge as well as their techniques to develop viable tissue or organs for patients [48]. To date, SD-TEVGs composed of a diverse array of materials ranging from synthetic to natural or a mixture hereof have been explored and eventually modified to examine their functionality as porous scaffold for supporting cell growth and tissue regeneration either in vitro or in vivo (Figure 1 and Table 8).

**Table 4 cells-10-00713-t004:** SD-TEVGs evaluated in large animals in arterial bypass using end-to-side (ETS) anastomosis.

Study Group	Model						Graft		Modification					Outcome		
	D (mm)	L (cm)	Animal	Implantation site	Anastomosis	Antithro-mbotic therapy	Graft type	Material	Chemical Modification	Biological modification = Recellularization	Luminal cell type	Medial cell type	Mechanical modification = Precondition	Follow-up (day)	Patency	Graft Failure
Mahara 2015 [49] 7 days control	2	25	Pig	Femorale-femoral artery crossover bypass	Proximal: STE Distal: ETE	No	Xenogeneic	Acellular ostrich carotid artery	None	None	None	None	No	7	0	Thrombus
Mahara 2015 [49] Peptide-modified	2	25	Pig	Femorale-femoral artery crossover bypass	Proximal: STE Distal: ETE	No	Xenogeneic	Acellular ostrich carotid artery	POG7G3REDV	None	None	None	No	20	83%	Unstable suturing at proximal anastomotic site
Fang 2019 [50] dHUA	4	4	Sheep	Carotid artery	ETS	No	Xenogeneic	Decellularized human umbilical artery	None	None	None	None	No	28	0	Thrombus
Fang 2019 [50] dSCA	4	4	Sheep	Carotid artery	ETS	No	Allogeneic	Decellularized sheep carotid artery	None	None	None	None	No	28	0	Distal stenosis
Fang [45] PCL w/o enoxaparin	4	4	Sheep	Carotid artery	ETS	No	Synthetic	PCL	None	None	None	None	No	28	0	Thrombus
Dahl 2011 [51] Dog coronary, 1 month	3 or 4	4–8.5	Dog	Coronary or carotid artery	ETS *	Yes	Autologous cells on allograft	Decellularized graft from allogeneic canine cells grow on a PGA scaffold	None	Autologous	Vessel-EC	None	Yes	7–365	83%	NR
Fang [45] PCL w enoxaparin	4	4	Sheep	Carotid artery	ETS	Yes	Synthetic	PCL	None	None	None	None	No	28	100%	No failure
Nakayama 2018 [52] Arterial bypass	4	25	Dog	Carotid artery	ETS	Yes	Allogeneic	Ethanol fixed IBTA	None	None	None	None	No	30	100%	No failure
Soldani 2010 [53] ePTFE 6 months	7	5	Sheep	Carotid artery	ETS + carotid resection	Yes	Synthetic	ePTFE	None	None	None	None	No	180	50%	Thrombus
Soldani 2010 [53] PEtU 24 months	7	5	Sheep	Carotid artery	Proximal: ETE Distal: ETS	Yes	Synthetic	PU	None	None	None	None	No	730	100%	No failure

D: diameter; L: length; ETS: end-to-side anastomosis; ETE: end-to-end anastomosis; STE: side-to-end anastomosis; dHUA: decellularized human umbilical artery; dSCA: decellularized sheep carotid artery; PCL: polycaprolactone; IBTA: in-body tissue architecture; ePTFE: expanded polytetrafluoroethylene; POG7G3REDV: heterobifunctional peptide: (Pro-Hyp-Gly)_7_-Gly-Gly-Gly)-Arg-Glu-Asp-Val (REDV); EC: endothelial cell; and NR: not reported. * Interpreted from figure.

**Table 5 cells-10-00713-t005:** SD-TEVGs evaluated in large animal in arterial bypass using end-to-end (ETE) anastomosis (without antithrombotic therapy).

Study Group	Model						Graft		Modification					Outcome		
	D (mm)	L (cm)	Animal	Implantation site	Anastomosis	Antithro-mbotic therapy	Graft type	Material	Chemical Modification	Biological modification = Recellularization	Luminal cell type	Medial cell type	Mechanical modification = Precondition	Follow-up (day)	Patency	Graft Failure
Aper 2016 [54] 1 month	5.6	9	Sheep	Carotid artery	ETE	No	Natural (xenogeneic fibrin)	Highly compacted human fibrin matrix	Factor XIII	Autologous	PB-EC	PB-SMC	No	30	33%	Rupture
Aper 2016 [54] 6 months	5.6	9	Sheep	Carotid artery	ETE	No	Natural (xenogeneic fibrin)	Highly compacted human fibrin matrix	Factor XIII	Autologous	PB-EC	PB-SMC	No	180	100%	No failure
Cho 2005 [55] Acellular control	3	4	Dog	Carotid artery	ETE	No	Allogeneic	Decellularized canine carotid arteries	None	None	None	None	No	14	0	Thrombus
Cho 2005 [55] BMC	3	4	Dog	Carotid artery	ETE	No	Autologous cells on allograft	Decellularized canine carotid arteries	None	Autologous	BMMNC-EC	BMMNC-SMC	No	56	33%	Thrombus
Dahan 2017 [46] Acellular control	4	4.5	Pig	Carotid artery	ETE	Not mentioned	Allogeneic	Decellularized porcine carotid artery	None	None	None	None	No	42	100%	Even patent but still very narrowed lumen according to the staining
Dahan 2017 [46] scaECM	4	4.5	Pig	Carotid artery	ETE	Not mentioned	Autologous cells on allograft	Decellularized porcine carotid artery	None	Autologous	Vein-EC	Artery-SMC	Yes	42	100%	No failure
He 2002 [56] Type A, 1 month	5	5	Dog	Carotid artery	ETE	No	Autologous cells on synthetic and natural graft	Autologous SMCs-inoculated bovine collagen gel layer and an EC monolayer wrapped with PU-nylon mesh	None	Autologous	Vein-EC	Vein-SMC	No	30	100%	No failure, but dilation/ delamination was seen
He 2002 [56] Type B, 6 months	5	5	Dog	Carotid artery	ETE	No	Autologous cells on synthetic and natural graft	Autologous SMCs-inoculated bovine collagen gel layer and an EC monolayer wrapped with an excimer laser-directed microporous SPU film	None	Autologous	Vein-EC	Vein-SMC	No	180	100%	No failure
He 2003 [57] 1 month	4.5	6	Dog	Carotid artery	ETE	No	Autologous cells on synthetic and natural graft	Bovine collagen type I meshes wrapped with a SPU thin film	None	Autologous	PB-EPCs	None	No	30	83%	Dilation and thrombus
He 2003 [57] 3 months	4.5	6	Dog	Carotid artery	ETE	No	Autologous cells on synthetic and natural graft	Bovine collagen type I meshes wrapped with a SPU thin film	None	Autologous	PB-EPCs	None	No	90	100%	No failure
Narita 2008 [58] Acellular DU control	3	4.5	Dog	Carotid artery	ETE	No	Allogeneic	Decellularized ureters	None	None	None	None	No	7	20%	NR
Narita 2008 [58] Acellular DU control	3	4.5	Dog	Carotid artery	ETE	No	Allogeneic	Decellularized ureters	None	None	None	None	No	56	20%	NR
Narita 2008 [58] DU + EC + myfibroblasts	3	4.5	Dog	Carotid artery	ETE	No	Autologous cells on allograft	Decellularized ureters	None	Autologous	Vein-EC	Myofibroblasts	No	168	100%	No failure
Narita 2008 [58] PTFE control	3	4.5	Dog	Carotid arterial	ETE	No	Synthetic	PTFE	None	None	None	None	No	7	0	NR
Scherner 2014 [59] BC tube	3.5	10	Sheep	Carotid artery	ETE	No	Microbiological derived	Bacterial cellulose	None	None	None	None	No	84	50%	Thrombus formation next to the proximal anastomosis
Weber 2017 [44] Non-anti-platelet control	4.5	10	Sheep	Carotid artery	ETE	No	Microbiological derived	Bacterial nanocellulose	None	None	None	None	No	56	0	NR
Ye 2012 [60] PCL + heparin	2	4	Dog	Femoral artery	ETE	No	Synthetic	PCL	Heparin	None	None	None	No	28	100%	No failure
Zhao 2010 [61] 2 months	3	4	Sheep	Carotid artery	ETE	No	Autologous cells on allograft	Decellularized ovine carotid artery	None	Autologous	MSCs differentiated ECs-like cells	MSCs differentiated SMCs-like cells	No	60	100%	No failure
Zhao 2010 [61] 5 months	3	4	Sheep	Carotid artery	ETE	No	Autologous cells on allograft	Decellularized ovine carotid artery	None	Autologous	MSCs differentiated ECs-like cells	MSCs differentiated SMCs-like cells	No	150	100%	No failure
Zhao 2010 [61] Acellular control	3	4	Sheep	Carotid artery	ETE	No	Allogeneic	Decellularized ovine carotid artery	None	None	None	None	No	14	0	Thrombus

ECM: extracellular matrix; ETE: end-to-end anastomosis; PB-EC: peripheral blood-endothelial cell; PB-SMC: peripheral blood-smooth muscle cell; BMMNC-EC: bone marrow mononuclear cells-endothelial cell; BMMNC-SMC: bone marrow mononuclear cells-smooth muscle cell; SMC: smooth muscle cell; EC: endothelial cell; EPCs: endothelial progenitor cells; MSCs: mesenchymal stem cells; PU: polyurethane; SPU: segmented polyurethane; PTFE: polytetrafluoroethylene; PCL: polycaprolactone; and NR: not reported.

**Table 6 cells-10-00713-t006:** SD-TEVGs evaluated in large animals in arterial bypass using end-to-end (ETE) anastomosis (with antithrombotic therapy).

Study Group	Model						Graft		Modification					Outcome		
	D (mm)	L (cm)	Animal	Implantation site	Anastomosis	Antithro-mbotic therapy	Graft type	Material	Chemical Modification	Biological modification = Recellularization	Luminal cell type	Medial cell type	Mechanical modification = Precondition	Follow-up (day)	Patency	Graft Failure
Arts 2002 [62] Transduction 3 weeks	4	5	Dog	Carotid artery	ETE	Yes	Synthetic	ePTFE	None	Autologous	Fat-derived microvascular endothelial cells	None	No	21	100%	No failure
Arts 2002 [62] 1 month	4	5	Dog	Carotid artery	ETE	Yes	Synthetic	ePTFE	None	Autologous	Fat-derived microvascular endothelial cells	None	No	30	83%	Thrombus
Arts 2002 [62] 1 month control	4	5	Dog	Carotid artery	ETE	Yes	Synthetic	ePTFE	None	None	None	None	No	30	83%	Thrombus
Arts 2002 [62] 12 months	4	5	Dog	Carotid artery	ETE	Yes	Synthetic	ePTFE	None	Autologous	Fat-derived microvascular endothelial cells	None	No	365	100%	No failure
Arts 2002 [62] 12 months control	4	5	Dog	Carotid artery	ETE	Yes	Synthetic	ePTFE	None	None	None	None	No	365	0	Organised thrombus
Chue 2004 [63] no mesh	3.75	6	Dog	Femoral artery	ETE	Yes	Autologous cells on autologous ECM	IBTA (from peritoneal and pleural cavities, based on Polyethylene or C-flex)	None	None	None	None	No	90–195	83%	Organized thrombus
Chue 2004 [63] PGA mesh	3.75	6	Dog	Femoral artery	ETE	Yes	Autologous cells on autologous ECM and synthetic graft	IBTA with biodegradable PGA mesh (from peritoneal and pleural cavities, based on Polyethylene)	None	None	None	None	No	90–195	75%	Organized thrombus
Chue 2004 [63] polypropylene mesh	3.75	6	Dog	Femoral artery	ETE	Yes	Autologous cells on autologous ECM and synthetic graft	IBTA with nonbiodegradable polypropylene mesh (from peritoneal and pleural cavities, based on Polyethylene)	None	None	None	None	No	90–195	0	Organized thrombus
Ju 2017 [64] Acellular control	4.75	5	Sheep	Carotid artery	ETE	Yes	Synthetic and natural	Bilayered blending of PCL and calf type I collagen	None	None	None	None	No	10	0	Thrombus
Ju 2017 [64] EC + SMC + flow	4.75	5	Sheep	Carotid artery	ETE	Yes	Autologous cell on synthetic and natural graft	Bilayered blending of PCL and calf type I collagen	None	Autologous	PB-EC	Artery-SMCs	Yes	180	100%	No failure
Kaushal 2001 [65] 130 days	4	4.5	Sheep	Carotid artery	ETE	Yes	Autologous cells on xenograft	Decellularized porcine iliac blood artery	None	Autologous	PB-EC	None	Yes	130	100%	No failure
Kaushal 2001 [65] Acellular control	4	4.5	Sheep	Carotid artery	ETE	Yes	Xenogeneic	Decellularized porcine iliac blood artery	None	None	None	None	No	15	25%	Thrombus
L’Heureux 1998 [66] w/o EC	3	5	Dog	Femoral artery	ETE (interpreted from figure)	Yes, immunosuppression	Xenogeneic	Dehydrated Human vascular SMC and fibroblasts cells sheet	None	None	None	None	No	7	50%	Thrombus
Ma 2017 [67] DAFP + EC	4 (outer diameter)	6	Dog	Carotid artery	ETE	Yes	Autologous cells on xenograft	Decellularized aortae of fetal pigs	None	Autologous	Vein-EC	None	Yes	180	100%	No failure
Mrowczynski 2014 [68] ePTFE control	4	5	Pig	Carotid artery	ETE	Yes	Synthetic	ePTFE	None	None	None	None	No	28	67%	NR
Mrowczynski 2014 [68] PCL	4	5	Pig	Carotid artery	ETE	Yes	Synthetic	PCL	None	None	None	None	No	28	78%	Thrombus from prosthetic kink
Neff 2011 [69] dsTEBV (EC + SMC)	5	6	Sheep	Carotid artery or femoral artery	ETE	Yes	Autologous cells on xenograft	Decellularized porcine carotid arterial segments	None	Autologous	PB-EC	Artery-SMC	Yes	120	100%	No failure
Neff 2011 [69] ecTEBV (EC)	5	6	Sheep	Carotid artery or femoral artery	ETE	Yes	Autologous cells on xenograft	Decellularized porcine carotid arterial segments	None	Autologous	PB-EC	None	Yes	120	100%	No failure
Nemcova 2001 [70] Acellular xenograft	4	5	Dog	Femoral artery	ETE	Yes	Xenogeneic	Acellular porcine small intestinal submucosa	Type I bovine collagen	None	None	None	No	63	89%	Wall thickening
Rothuizen 2016 [71] Tissue capsule	4.2	4	Pig	Carotid artery	ETE	Yes	Autologous ECM and synthetic graft	IBTA (from subcutaneous, based on PEOT/PBT)	None	None	None	None	No	28	88%	Peri-anastomotic intimal hyperplasia
Turner 2006 [72] Collagen coating	4.5	4.5	Goat	Carotid artery	ETE	Yes	Allogeneic cells on synthetic graft	PU	Alpha-2(VIII) collagen	Allogeneic	Artery-ECs	None	No	1	100%	No failure
Turner 2006 [72] Fibronectin coating	4.5	4.5	Goat	Carotid artery	ETE	Yes	Allogeneic cells on synthetic graft	PU	Fibronectin	Allogeneic	Artery-ECs	None	No	1	100%	No failure
Turner 2006 [72] Uncoated control	4.5	4.5	Goat	Carotid artery	ETE	Yes	Allogeneic cells on synthetic graft	PU	None	Allogeneic	Artery-ECs	None	No	1	0	Occlusive red thrombus developed from distal white thrombus
Wang 2019 [73] IBTA 2 months	3.9	6	Pig	Carotid artery	ETE	Yes	Autologous ECM	Decellularized IBTA (from subcutaneous, based on PTFE)	Heparin	None	None	None	No	60	67%	Anastomotic stenosis resulting thrombus
Weber 2017 [44] DAT, 9 months	4.5	10	Sheep	Carotid artery	ETE	Yes	Microbiological Derived	Bacterial nanocellulose	None	None	None	None	No	270	67%	Thrombus formation next to the proximal anastomosis
Weber 2017 [44] Smooth + DAT, 2 months	4.5	10	Sheep	Carotid artery	ETE	Yes	Microbiological derived	Bacterial nanocellulose	None	None	None	None	No	60	80%	NR
Wulff 2017 [74] SSHS-coated ePTFE	3.5	20	Sheep	Carotid artery	Interposition	Yes	Synthetic	ePTFE	Semisynthetic heparan sulphate-like on SEPS layer	None	None	None	No	140	25%	Anastomotic neointimal hyperplasia originating from the genuine vessel + delamination SEPS delaminated from the ePTFE graft
Wulff 2017 [74] Uncoated ePTFE control	3.5	20	Sheep	Carotid artery	Interposition	Yes	Synthetic	ePTFE	None	None	None	None	No	140	13%	Anastomotic neointimal hyperplasia originating from the genuine vessel
Zhou 2009 [75] DS control	3	4.5	Dog	Carotid artery	ETE	Yes	Allogeneic	Decellularized canine carotid arteries	None	None	None	None	No	180	47%	Thrombus
Zhou 2009 [75] VHDS graft	3	4.5	Dog	Carotid artery	ETE	Yes	Allogeneic	Decellularized canine carotid arteries	Heparin and VEGF	None	None	None	No	180	93%	Thrombus
Zhou 2012 [76] Acellular control (DV)	3	4.5	Dog	Carotid artery	ETE	Yes	Allogeneic	Decellularized canine carotid arteries	None	None	None	None	No	90	60%	Thrombus
Zhou 2012 [76] Heparin + EPC	3	4.5	Dog	Carotid artery	ETE	Yes	Autologous cells on allograft	Decellularized canine carotid arteries	Heparin	Autologous	PB-EC	None	Yes	90	95%	Thrombus
Zhou 2014 [77] Acellular control	3	4.5	Dog	Carotid arteries	ETE	Yes	Synthetic and natural graft	CS/PCL	None	None	None	None	No	90	17%	Thrombus
Zhou 2014 [77] CS/PCL + OEC	3	4.5	Dog	Carotid artery	ETE	Yes	Autologous cells on synthetic and natural graft	CS/PCL	None	Autologous	PB-EC	None	Yes	90	83%	Thrombus

ETE: end-to-end anastomosis; ECM: extracellular matrix; PTFE: polytetrafluoroethylene; ePTFE: expanded polytetrafluoroethylene; IBTA: in-body tissue architecture; C-flex: styrene-ethylenebutylene modified block co-polymer with silicone oil; PGA: polyglycolic acid; PCL: polycaprolactone; PEOT/PBT: poly (ethylene oxide terephthalate) epoly (butylene terephthalate); SMC: smooth muscle cell; PU: polyurethane; CS/PCL: Chitosan/poly(e-caprolactone); SEPS: styrene ethylene propylene styrene co-polymer; VEGF: vascular endothelial growth factor; PB-EC: peripheral blood-endothelial cell; and EC: endothelial cell.

**Table 7 cells-10-00713-t007:** SD-TEVGs evaluated in large animal in arteriovenous shunt.

Study Group	Model						Graft		Modification					Outcome		
	D (mm)	L (cm)	Animal	Implantation site	Anastomosis	Antithro-mbotic therapy	Graft type	Material	Chemical Modification	Biological modification = Recellularization	Luminal cell type	Medial cell type	Mechanical modification =Precondition	Follow-up (day)	Patency	Graft Failure
Koenneker 2010 [78] 6 months control	5 or 6	7.5	Sheep	Cervical AV shunts	ETS	No	Xenogeneic	Decellularized bovine internal thoracic arteries	None	None	None	None	No	180	71%	NR
Koenneker 2010 [78] 6 months	5 or 6	7.5	Sheep	Cervical AV shunts	ETS	No	Autologous cells on xenograft	Decellularized bovine internal thoracic arteries	None	Autologous	PB-EC	None	Yes	180	86%	NR
Koenneker 2010 [78] 3 months control	5 or 6	7.5	Sheep	Cervical AV shunts	ETS	No	Xenogeneic	Decellularized bovine internal thoracic arteries	None	None	None	None	No	90	83%	NR
Koenneker 2010 [78] 3 months	5 or 6	7.5	Sheep	Cervical AV shunts	ETS	No	Autologous cells on xenograft	Decellularized bovine internal thoracic arteries	None	Autologous	PB-EC	None	Yes	90	100%	No failure
Syedain 2017 [79] BAVG in general	4	12.5	Baboon	Axillary-cephalic or axillarybrachial upper arm, AV shunt	ETS	Yes	Xenogeneic from two origins	Decellularized graft from human fibroblasts and bovine fibrin gel	None	None	None	None	Yes	180	45%	Unexplained rapid, occlusive thrombosis in five cases and rupture in one case
Dahl 2011 [51] Baboon, 6 months	6	12.5	Baboon	Axillary artery and the distal brachial vein	ETS (interpreted from figure)	Yes	Xenogeneic	Decellularized graft from human cells on a polymer scaffold	None	None	None	None	No	180	100%	No failure
Rotmans 2005 [80] 4 weeks control	5	7	Pig	Carotid artery and internal jugular vein	ETS	Yes	Synthetic	ePTFE	None	None	None	None	No	28	67%	Recent thrombotic occlusion on top of extensive IH in the venous outflow tract
Rotmans 2005 [80] 4 weeks	5	7	Pig	Carotid artery and internal jugular vein	ETS	Yes	Synthetic	ePTFE	Anti–human CD34 monoclonal antibodies	None	None	None	No	28	67%	Recent thrombotic occlusion on top of extensive IH in the venous outflow tract
Tillman 2012 [81] Late, 6 months	5	6	Sheep	Carotid artery to jugular vein	ETS	Yes	Autologous cells on xenograft	Decellularized porcine carotid artery	None	Autologous	PB-EC	None	Yes	168	0	Outflow stenosis from intimal hyperplasia at the venous anastomosis
Tillman 2012 [81] Early, 2 months	5	6	Sheep	Carotid artery to jugular vein	ETS	Yes	Autologous cells on xenograft	Decellularized porcine carotid artery	None	Autologous	PB-EC	None	Yes	60	71%	Thrombus due to kinking at the graft apex/Not identified
Li 2005 [47] 4 weeks control	5	6	Sheep	Femoral artery and vein or the carotid artery and jugular vein	Not mentioned	Not mentioned	Synthetic	ePTFE	None	None	None	None	No	28	100%	No failure
Li 2005 [47] 4 weeks	5	6	Sheep	Femoral artery and vein or the carotid artery and jugular vein	Not mentioned	Not mentioned	Synthetic	ePTFE	P15 cell-binding peptide	None	None	None	No	28	100%	No failure
Ong 2017 [82] ePTFE control	5	5	Sheep	Carotid artery to external jugular vein	ETS	Yes	Synthetic	ePTFE	None	None	None	None	No	28	100%	No failure
Ong 2017 [82] Nanofiber TEVG	5	5	Sheep	Carotid artery to external jugular vein	ETS	Yes	Synthetic	PGA/PLCL	None	None	None	None	No	28	67%	NR
Furukoshi 2019 [83] Slit patterns with straight or spiral lines	4	5	Dog	Common carotid artery and the jugular vein	Proximal: STS Distal: ETE	Yes	Autologous cells on autologous ECM	IBTA (from subcutaneous, based on silicone/steel)	None	None	None	None	No	28	100%	No failure
Furukoshi 2019 [83] Slit patterns with straight or spiral lines	4	5	Dog	Common carotid artery and the jugular vein	STS	Yes	Autologous cells on autologous ECM	IBTA (from subcutaneous, based on silicone/steel)	None	None	None	None	No	28	100%	No failure
Furukoshi 2019 [83] Slit patterns with straight or spiral lines	4	5	Dog	Common carotid artery and the jugular vein	Proximal: STE Distal: ETS	Yes	Autologous cells on autologous ECM	IBTA (from subcutaneous, based on silicone/steel)	None	None	None	None	No	28	100%	No failure
Nakayama 2018 [52] AV shunt	5	50	Goat	Carotid artery and jugular vein	ETS	Yes	Allogeneic ECM	Ethanol fixed IBTA (from subcutaneous, based on nylon)	None	None	None	None	No	30	100%	No failure
AV shunt: arteriovenous shunt; ETS: end-to-side anastomosis; STS: side-to-side anastomosis; ETE: end-to-end anastomosis; STE: side-to-end anastomosis; ECM: extracellular matrix; ePTFE: expanded polytetrafluoroethylene; PGA/PLCL: polyglycolic acid/poly(L-lactide-co-ε-caprolactone); IBTA: in-body tissue architecture; P15 cell-binding peptide: large cell-binding peptide consisting of 15 amino acids, Gly-Thr-Pro-Gly-Pro-Gln-Gly-IIe-Ala-Gly-Gln-Arg-Gly-Val-Val, PB-EC: peripheral blood-endothelial cell; IH: intimal hyperplasia; and NR: not reported.																

**Table 8 cells-10-00713-t008:** Fabrication methods of SD-TEVG.

Graft Type	Material	Fabrication	Reference
**Synthetic**	Dacron, ePTFE, PU, PCL, PLCL, PGA, PLA, PLLA, PLGA, PGS, PEUU	Electrospinning, molding, 3D Printing, laser degradation, hydrogel	[45,47,53,58,60,62,68,72,74,80,82]
**Natural**	Collagen, elastin, fibrin, hyaluronic acid, silk fibroin, gelatin, chitosan	Electrospinning, molding, rolling, 3D Printing, laser degradation, hydrogel	[44,54,59]
	Cell-secreted ECM	Hydrogel, rolling, self-assembled cell sheets	[66,79]
	Native ECM	Decellularization or crosslinking of native tubular organs (vessels, ureters and small intestinal submucosa)	[46,49,50,55,58,61,65,67,69,70,75,76,78,81]
**Hybrid**	Combination of above	Combination of above, e.g., in-body tissue architecture (IBTA)/ in vivo tissue engineering	[51,52,56,57,63,64,71,73,77,83]

Dacron: Polyethylene terephthalate (PET); ePTFE: expanded polytetrafluoroethylene; PU: Polyurethane; PCL: polycaprolactone; PLCL: poly(L-lactide-co-ε-caprolactone); PGA: polyglycolic acid; PLA: poly-lactic acid; PLLA: poly-l-lactic acid; PLGA: poly (lactide-co-glycolide); PGS: Poly(glycerol-sebacate); PEUU: poly(ether urethane urea); and ECM: extracellular cellular matrix.

#### 3.2.1. Synthetic SD-TEVGs

Synthetic SD-TEVGs have been extensively tested likely due to their easy availability and customization (Table 9).

Conventional prosthetic vascular grafts like Dacron and expanded polytetrafluoroethylene (ePTFE) are biostable grafts, which can prevent mechanical deformation in high-pressure and high-flow-rate arteries, like the aorta. Thus, they are safe grafts to be used in large diameter bypass grafting and offer satisfying patency rates and outcomes. However, they fail when applied in small-diameter bypass grafting, since they result in thrombus formation and do not facilitate endothelialization when uncoated [84]. They are also very rigid and do not match the compliance of the native vessel [85,86,87]. Considering the safe and positive experience when applying ePTFE as implants in patients, ePTFE therefore often serve as negative controls when evaluating other novel synthetic SD-TEVGs. In the studies screened by our systematic search (Table 4, Table 5, Table 6 and Table 7), 11 out of 39 studies have examined synthetic SD-TEVGs. Eight studies included SD-TEVGs made of pure ePTFE without any modification as a negative control group, four of which served as control for other four types of grafts (Table 4, Table 5, Table 6 and Table 7) [53,58,68,82] and four of which served as a scaffold for testing the effect of surface modifications (Table 6 and Table 7) [47,62,74,80]. Pure ePTFE grafts displayed a patency of 67–100% after about 1 month in five studies [47,62,68,80,82]. Four of these studies used systemic antithrombotic treatment, while one study did not refer to such treatment [47]. When the follow-up period increased to 140–365 days, the patency of ePTFE decrease to 0–50% even in the presence of systemic antithrombotic treatment [53,62,74]. The only study that examined pure ePTFE without systemic antithrombotic treatment revealed poor patency (0%) after 7 days [58]. Thus, pure ePTFE is inapplicable as a material for SD-TEVGs due to its poor patency resulting from high thrombogenicity [53] or late intimal hyperplasia [74,80]. These events are well known causes for graft failure and may reflect resistance of a hydrophobic surface to the endothelial cell (ECs), compliance mismatch, or prolonged foreign body reaction to the polymer [86,88,89,90].

Polyurethane (PU), another synthetic material based on advanced material techniques, has been produced as both biostable and biodegradable grafts [91] with compliance matching the native vessel [92]. Since biodegradable PU grafts show better outcomes, biodegradable synthetic polymers have attracted more and more attention in the field and are extensively investigated for developing SD-TEVGs. Recently explored biodegradable synthetic polymers include, but are not limited to, polycaprolactone (PCL), poly(L-lactide-co-ε-caprolactone) (PLCL), polyglycolic acid (PGA), poly-lactic acid (PLA), poly-l-lactic acid (PLLA), poly (lactide-co-glycolide (PLGA), poly(glycerol-sebacate) (PGS), and poly(ether urethane urea) (PEUU) [93,94,95,96,97,98,99]. Biodegradable synthetic vascular grafts have already been investigated in humans, and the first trial comprised a TEVG (diameter: 10 mm) consisting of PCL and PLA copolymers strengthened by woven PGA, which was then seeded with autologous venous patient cells and implanted into a 4-year-old patient for pulmonary-artery reconstruction by Shin’oka et al. in 2001 [100]. The same group further evaluated similar TEVGs seeded with bone marrow cells as large diameter conduits in patients with congenital heart disease for midterm and long-term functionality in 2005 and 2010 [101,102]. Likewise, PCL-based large-diameter vessel grafts have been implanted into patients with congenital heart disease [103], whereas PLLA has been tested for right ventricular outflow tract (RVOT) reconstruction [104]. However, all these clinical trials were conducted in large-diameter vessel sites [100,101,102,103], and still no small-diameter biodegradable synthetic vascular graft has been tested in humans, despite that the idea of using biodegradable material to support tissue reconstruction was presented decades ago [105]. By contrast, the retrieved large-animal studies reveal that several biodegradable synthetic polymers, including PU [53,72], PGA/PLCL [82] and PCL [45,60,68], have been examined as small-diameter vessel grafts in large-animal studies with clinically relevant length/diameter ratios. In this regard, Turner et al. reported that, already within 24 h, PU grafts were prone to occlusive red thrombus, developed from distal white thrombus, although the graft was endothelialized before implantation into goat carotid artery [72]. The authors noted only an 8% retention of seeded ECs on the PU grafts after exposure to high blood flow in vivo, which indicates a high thrombogenicity of PU grafts. However, the ECs seeded were from allogeneic origin, which may harm the retention of cells and in the end lower the patency. As compared to the synthetic PU polymer, natural alpha-2 (VIII) collagen or fibronectin polymer displayed enhancement in affinity for ECs when coated to the PU graft, with about 50% retention of ECs after exposure to high blood flow in vivo [72]. Interestingly, Soldani et al. also evaluated a PU graft composed of poly(ether)urethane–polydimethylsiloxane (PEtU–PDMS), a polymer belonging to the large PU family, and reported an excellent long-term patency at 100% after 2 years in sheep carotid bypass with 1 month postoperation antithrombotic treatment using aspirin [53], which is reported to fail in inhibiting platelet aggregation in sheep [106,107]. Thus, these two PU grafts display converse in vivo outcome. Besides PU, another polymer: PGA/PLCL has been involved in one retrieved study (Table 7). Ong et al. implanted PGA/PLCL grafts in an arteriovenous shunt model and presented 67% patency under systemic antithrombotic treatment after 28 days, which is lower than what has been observed in ePTFE controls (100%) [82]. As compared to other polymers, studies have shown favorable mechanical properties, biocompatibility, and appropriate biodegradability of PCL-based SD-TEVGs [88,108,109]. In agreement, we have recently shown that PCL scaffolds may be easily recellularized with adipose-derived regenerative cells (ADRCs) during in vitro static culture in a dish, as well as in dynamic culture using a 3D bioreactor with pulsatile media flow [45]. Specifically, we found vigorous proliferation of ADRCs in the PCL material in combination with complete ADRC migration throughout the scaffold under both static and dynamic conditions. These findings thus underscore the high cytocompatibility of a PCL-based SD-TEVGs. However, Mrowczynski et al. (Table 6) have previously reported a transprosthetic blood leakage of PCL grafts shortly after reconstruction of blood flow in bypass, whereas a late thrombotic occlusion resulted from a prosthetic kink [68]. To avoid such in vivo complications, we designed our PCL graft with a lower porosity (60 versus 80%) and a reduced thickness (250 versus 500 µm) [45,68]. Indeed, our in vivo results, when testing the PCL-based SD-TEVGs as sheep carotid substitution, suggested that transprosthetic blood leakage and prosthetic kink were successfully prevented [45]. Moreover, our PCL vessel graft (Table 4) displayed easy surgical handling, secured anastomosis, and absence of any rupture, aneurysm, or necrosis within the follow-up period, which further emphasizes its appropriate and stable mechanical properties and biodegradation. It is likely that our PCL is prevented from excessive degradation that would otherwise weaken the mechanical strength and lead to graft rupture or dilation [110]. Besides Mrowczynski’s [68] and our study [45], one other study (Table 4, Table 5 and Table 6) has investigated PCL in arterial bypass substitution [60]. Whereas Ye et al. conjugated heparin to the PCL graft and revealed a 100% patency after 4 weeks, Mrowczynski et al. demonstrated a 78% patency for PCL grafts without any conjugation but under systemic antithrombotic treatment after 4 weeks [60,68]. Thus, PCL-based grafts may maintain patency in a short period either under local conjugation or systemic administration of antithrombotic treatment. This is clearly supported by our study, showing that daily use of antithrombotic treatment increases graft patency [45]. Thus, similar to the PCL grafts in Mrowczynski and Ye’s studies [60,68], our PCL grafts (Table 4) exhibited 100% patency after 28 days with the daily use of antithrombotic treatment, whereas the PCL grafts without antithrombotic treatment displayed very poor patency (0%) already in the first week after implantation [45]. Thus, it seems that PCL-based SD-TEVGs are highly thrombogenic, but this may be significantly reduced by systemic antithrombotic treatment and recellularization as discussed below.

In general, the biodegradable synthetic SD-TEVGs evaluated in vivo in large-animal models manifest robust and stable mechanical properties required for in vivo hemodynamics. However, very limited types of synthetic SD-TEVGs have come to the large-animal evaluation, and their thrombogenicity and biocompatibility require substantial improvement through either antithrombotic treatment or modification on the grafts. Moreover, considering most studies included in the large-animal studies only reported short-term (<28 days) results, longer follow-up in vivo is valuable and thus needed for understanding the medium- and long-term outcome and interaction mechanisms between this polymer and host environment.

#### 3.2.2. Natural SD-TEVGs

Another popular type of materials are those polymers and matrix naturally present in humans and animals or generated by cells or even microorganisms (Figure 1). Natural SD-TEVGs manifest low immunogenicity, high biocompatibility, and an inherent biodegradability, which suppress the immune response and facilitate cell growth and tissue regeneration (Table 9). Nineteen out of 39 retrieved large-animal studies (Table 4, Table 5, Table 6 and Table 7) have evaluated natural SD-TEVGs. Three out of these 19 studies have used natural polymer-based SD-TEVGs [44,54,59], whereas two studies investigated cell-secreted extracellular cellular matrix (ECM)-based SD-TEVGs [66,79], and the 14 remaining used native ECM-based SD-TEVGs (Table 8).

#### 3.2.3. Natural Polymer-Based SD-TEVGs

Natural polymers including collagen, elastin, fibrin, hyaluronic acid, silk fibroin, gelatin, and chitosan [111] can be artificially constructed into TEVGs by fabrication methods like electrospinning, molding, rolling, 3D printing, laser degradation, and hydrogel (Figure 1 and Table 8). Collagen and elastin are especially interesting molecules [112,113], which are components of the ECM in native blood vessels. In this regard, Aper et al. reported a SD-TEVG, based on a highly compact fibrin matrix, with a 100% patency after 180 days when the surface was coated with Factor XIII and recellularized by both ECs and smooth muscle cell (SMCs) before implantation [54]. However, graft rupture, which is a typical failure due to weak mechanical strength and fast natural polymer degradation, was noted in the 30 day observation group. Scherner’s group developed another type of SD-TEVGs with natural polymers, composed of a microbiological derived matrix called bacterial nanocellulose (BNC). They developed a first [59] and second [44] generation, where the main difference concerned the wall thickness (2.0–3.5 mm versus 1.0–2.5 mm, respectively). First-generation BNCs displayed no graft rupture due to their high mechanical strength, but 50% failed due to thrombus formation next to the proximal anastomosis after 84 days without any thrombotic treatment [59]. By contrast, second-generation BNC grafts demonstrated 0% patency after 56 days without dual antiplatelet treatment [44], thus even worse than the first-generation BNC grafts. However, the patency of second-generation BNC grafts may be rescued by dual antiplatelet treatment (67% patency after 270 days) and surface smoothing (80% patency after 60 days) as shown by Weber et al. [44].

#### 3.2.4. Cell-Secreted ECM-Based SD-TEVGs

Like natural polymer-based SD-TEVGs, cell-secreted ECM-based SD-TEVGs are constructed by a similar fabrication method but involve cultivation of cells (Figure 1 and Table 8). L’Heureux et al. have generated a three-layered graft consisting of human fibroblast and SMC sheets, which has a mechanical strength close to that found in human vessels [66]. When the grafts were dehydrated and evaluated in large-animal models, the patency was 50% after 7 days with grafts failing from thrombus formation [66]. These multisheeted grafts are fabricated by rolling of self-assembled cell sheets where the resulting vascular graft consists of ECM produced by in vitro cultured cells, with the nuclei and cell cytoplasm being removed afterwards by dehydration [33,66]. Indeed, L’Heureux’s group have evaluated such grafts as arteriovenous shunts in human (Table 2) as discussed above [31,33,36,37]. Likewise, by combining both a natural polymer (bovine fibrin gel) and cell-secreted ECM from human fibroblasts, Syedain et al. have generated a natural SD-TEVG that was further preconditioned in a pulsed flow-stretch bioreactor before decellularization. Upon in vivo in baboon, these grafts displayed 45% patency after 180 days as arteriovenous grafts, where the low patency was caused by graft rupture and thrombus formation [79].

#### 3.2.5. Native ECM-Based SD-TEVGs

Finally, native ECM can be obtained from native vessels or other tubular organs from animals and humans to fabricate natural SD-TEVGs by using crosslinking or decellularization procedures (Figure 1 and Table 8) [114]. By these procedures, the ECM and its structure are preserved, and the immunogenicity of native tissues may be eliminated before implantation [115]. Compared to crosslinking, decellularization has gained massive attention in the past decade. Firstly, decellularization removes the immunogenic genetic material in tissue through chemical, enzymatic, or physical approaches [116]. For example, detergents like sodium dodecyl sulfate (SDS) or Triton X-100 and enzymes like DNAse are effective in removing proteins and nucleic acids whereby the immunogenicity is reduced [115]. Secondly, decellularization, if carefully adjusted, preserves the native ECM and their ultrastructure that are thought to benefit stem cell migration into the desired scaffold region and stimulate correct differentiation into functional cell types like endothelial cells and SMCs. Several clinical trials have already shown biocompatibility of decellularized vessels for use in dialysis, heart valves, and vaginal organs [34,117,118,119,120,121].

In the identified large-animal studies (Table 4, Table 5, Table 6 and Table 7), 14 studies have evaluated native ECM-based SD-TEVGs, all based on decellularization rather than crosslinking. The involved native organs include carotid artery (dog [55,75,76], pig [46,69], sheep [50,61], ostrich [49]), iliac artery (pig [65]), fetal aortae (pig [67]), umbilical artery (human [50]), ureters (dog [58]), and small intestinal submucosa (pig [70]) as arterial bypass grafts, whereas internal thoracic artery (cow [78]) and carotid artery (pig [81]) have been used as grafts for arteriovenous shunts. These SD-TEVGs range in diameter from 3 to 6 mm. In arterial bypass models, most xenogeneic native ECM-based SD-TEVGs have turned out to fail due to thrombus within short time (<15 days), when no modification like chemical modification or recellularization have been applied to the surface of SD-TEVGs [49,50,65]. Decellularized ostrich carotid artery and human umbilical artery grafts display early occlusion (<7 days) from thrombus in xenogeneic sheep models [49,50]. By inclusion of antithrombotic treatment, the early thrombotic occlusion (<7 days) of xenogeneic decellularized human umbilical artery may be prevented (unpublished data from our lab). Similarly, Nemcova et al. have evaluated decellularized xenogeneic SD-TEVGs composed of porcine small-intestinal submucosa coated with bovine type I collagen under antithrombotic treatment and obtained a patency of 89% after 63 days in dog. In vivo failure occurred at 8 weeks due to wall thickening instead of thrombotic occlusion [70]. However, despite the use of antithrombotic treatment, decellularized xenogeneic porcine iliac artery demonstrates only 25% patency after 15 days due to thrombotic occlusion [65]. Similar to xenogeneic decellularized SD-TEVGs, decellularized allogeneic SD-TEVGs are prone to occlusion as arterial bypass graft without antithrombotic treatment [46,50,55,58,61]. Narita et al. estimated that decellularized allogeneic ureters vessel grafts have a 20% patency after 7 and 56 days, which is similar to that of synthetic ePTFE [58]. Moreover, a similar patency (0% after 14 days [61] and 0% after 28 days [50]) have been reported for decellularized allogeneic sheep carotid artery. The failure of decellularized allogeneic grafts seems to occur by several mechanisms. Zhao et al. have observed early thrombus in the decellularized sheep carotid artery (dSCA) within 2 weeks [61], whereas we observed intimal hyperplasia rather than thrombus in dSCA after 2–4 weeks [50]. The absence of early thrombus in allogeneic dSCA in our study could be explained by better mechanical match between graft and host artery, since our grafts were harvested from adult sheep and implanted into sheep of the same age, while grafts in Zhao’s study were harvested from 6- to 8-month-old sheep and implanted into 12-month-old sheep. However, it is also notable that the bodyweight (57.3–75.6 kg) of sheep used in our study is much higher than that (20–25 kg) used in Zhao’s study. The overall diameter of both the dSCA grafts and the host carotid arteries in our study are therefore larger than that in Zhao’s study, which may also contribute to the absence of early thrombus in our dSCA grafts [122]. Moreover, without antithrombotic treatment, the decellularized allogeneic canine carotid artery displays a similar poor patency (0% after 14 days) due to thrombus as reported by Cho et al. [55]. The patency of decellularized allogeneic grafts indeed seems to be improved by antithrombotic treatment [75,76]. As such, Zhou et al. have shown that decellularized allogeneic canine carotid artery exhibit a patency at 47% after 180 days and at 60% after 90 days [75,76], which is substantially higher than the patency for decellularized allogeneic grafts without antithrombotic treatment [50,55,58,61]. As also seen with other types of SD-TEVGs, grafts display a higher patency as arteriovenous shunt than as arterial bypass. Accordingly, decellularized xenogeneic bovine internal thoracic arteries used for arteriovenous shunts display 83 and 71% patency after 90 and 180 days, respectively. The patency was even increased to 100 and 86% with autologous ECs recellularization [78], which is known in general to improve graft patency [122]. Decellularized xenogeneic porcine carotid arteries with autologous EC recellularization also exhibit relative high mid-term patency (71% after 60 days), which is however substantially deteriorated (0% after 168 days) likely due to intimal hyperplasia induced outflow stenosis at the venous anastomosis [81].

Thus, the large-animal studies clearly reveal that, natural polymer- and cell-secreted ECM-based SD-TEVGs tested in large animals still fail due to a weak strength and thrombogenicity [44,54,59,66,79], whereas decellularized native ECM-based SD-TEVGs possess adequate mechanical strength, since rupture is rarely reported, but their patency remains depending on their origin (xenogeneic or allogeneic) (Table 9). Furthermore, antithrombotic treatment and modifications to the decellularized scaffold such as heparin, or POG7G3REDV peptide conjugation [49,75] or recellularization with autologous cells including their precondition [58,61,65,69,76,78] may significantly improve the patency of decellularized native ECM-based SD-TEVGs. However, availability of the native organ source needs to be taken into consideration (Table 9). For instance, arteries from a donor person as exemplified by the carotid artery, iliac artery and internal thoracic artery, tubular organs like ureters and small intestinal submucosa, or fetal tissue like fetal aortae remain scarce in numbers since their isolation requires an available donor and always includes a risk for the donor. Making donor banks from diseased persons would offer an alternative, but the age and associated diseases should be considered. In contrast to this, human umbilical artery is a waste product globally and is thus representing an abundant source without ethical issue.

#### 3.2.6. Hybrid SD-TEVGs

Hybrid TEVGs refers to vascular grafts fabricated by combining synthetic and natural molecules or matrix aiming at better performance, since they may bring the advantages of mechanical strength, high biocompatibility, and availability in large-scale production (Table 9). As an example, Nagiah et al. has developed highly compliant SD-TEVGs using synthetic polymers of PU, PCL, or PLA sheathed by gelatine [123]. Not only using a mixture or combination of synthetic and natural materials but also a combination of fabrication methods is getting more and more popular to resolve the challenges with generation of a high complex structure like a small-diameter blood vessel. In-body tissue architecture (IBTA) remains another example, which is also known as in vivo tissue engineering [13]. In 1999, Campbell et al. pioneered this idea by implanting a silastic tube into the peritoneal or pleural cavity of dogs to fabricate autologous SD-TEVGs [63,124]. The rationale for this method to be suggested, is that it implicates the anticoagulant ability of mesothelial cells [125] and the foreign body reaction to biomaterials [126]. Nowadays, this idea is still being explored though with biodegradable synthetic material being implanted for development within a subcutaneous cavity instead of the peritoneal or pleural cavity [73,83,127,128]. As for the large-animal studies (Table 4, Table 5, Table 6 and Table 7), 10 studies have evaluated hybrid SD-TEVGs as arterial bypass grafts [51,52,56,57,63,64,71,73,77,83] and arteriovenous shunts [51,52,83].

#### 3.2.7. In-Vitro-Developed Hybrid SD-TEVGs

Five of the above 10 mentioned studies evaluated the hybrid SD-TEVGs constructed in vitro by blending polymer or cells (Figure 1) [51,56,57,64,77]. As such, Dahl et al. engineered SD-TEVGs by growing allogeneic SMCs on rapidly degraded PGA tubular grafts followed by decellularization [51]. The hybrid SD-TEVGs were further seeded with autologous ECs and evaluated in both carotid and coronary-artery bypass. Although a very distinguished coverage (0–60%) of ECs was achieved, most of the grafts (83%) maintained patency after a follow-up from 7 to 365 days, which may both be attributed to systematic antithrombotic treatment and also the small numbers of residing ECs. Similarly, He et al. has grown autologous SMCs in a type I collagen gel which was then wrapped in segmented PU film as an outer layer. The SD-TEVGs were seeded with autologous ECs and displayed 100% patency after 180 days in canine carotid artery bypass with confluent coverage of ECs, even without applying systemic antithrombotic treatment [56]. A similar hybrid graft composed of collagen I and wrapping segmented PU film manifested similar patency (100% after 90 days) in vivo although SMCs were not blended in during fabrications, which suggests that the mechanical strength of these segmented PU-based hybrid SD-TEVGs derived mainly from the PU film rather than the SMCs and the secreted ECM from cells [57]. Aside from PU-based hybrid SD-TEVGs, two studies have developed PCL-based hybrid SD-TEVGs. As such, Zhou et al. developed hybrid SD-TEVGs by blending chitosan with PCL and found a poor patency of 17% after 90 days [77], and similarly Ju et al. reported zero patency for hybrid SD-TEVGs composed by type I collagen and PCL after 10 days [64], both as carotid arterial bypass grafts under antithrombotic treatment [64,77]. This contrasts the relative high patency of PCL synthetic SD-TEVGs under antithrombotic treatment observed by Mrowczynski [68] and us [45] and might be explained by the incorporation of xenogeneic natural polymers that are known to accelerate blood clotting, like collagen [64] or chitosan [77], into the SD-TEVGs. In specific, type I collagen is the dominant collagen in human arteries and a strong trigger for platelet aggregation and thrombosis [129], and similarly chitosan has been shown to have hemostatic effects [130]. Even so, a significant increased patency (100% after 100 days) was seen when preconditioned autologous ECs and SMCs were seeded on type I collagen/PCL hybrid SD-TEVGs [64], and likewise an improved patency (83% after 90 days) was reported for autologous ECs seeded chitosan/PCL hybrid SD-TEVGs [77]. Thus, to further evaluate the contribution to in vivo patency from the improved affinity between ECs and the natural type I collagen or chitosan polymers of PCL-based hybrid SD-TEVGs, more large-animal studies evaluating endothelialized PCL-based hybrid SD-TEVGs are needed. Neverthless, in-vitro-developed hybrid SD-TEVGs indeed seem to have favorable patency when seeded with endothelial cells [51,56,57,64,77]. However, as unseeded hybrid grafts [64,77], they exhibit very poor patency despite the use of antithrombotic treatment, likely due to the incorporation of xenogeneic natural polymers as discussed above. This is similar to what has been observed for xenogeneic native ECM-based SD-TEVGs [65], indicating that the xenogeneic originated molecules may harm the graft outcome (Table 9), but this can be rescued by endothelialization for an intermediate period.

#### 3.2.8. IBTA-Based SD-TEVGs

As compared to the above mentioned in-vitro-developed hybrid SD-TEVGs, IBTA-based SD-TEVGs utilize the host environment as an in vivo bioreactor to initiate the foreign body reaction against synthetic tubing mold and engineer ECM (Figure 1) [126]. The low immunogenicity of IBTA-based SD-TEVGs is a clear advantage, but the off-shelf time and unavailability for acute procedures are disadvantages (Table 9). Currently, five studies have evaluated IBTA-based SD-TEVGs in vivo in large animals (Table 4, Table 5, Table 6 and Table 7), and all these studies applied the use of systemic antithrombotic treatment [52,63,71,73,83]. For instance, Chue et al. inserted polyethylene tubing, either bare or wrapped in a biodegradable PGA mesh or a nonbiodegradable polypropylene (Prolene) mesh, in the peritoneal or pleural cavity of dogs for three weeks and then harvested autologous IBTA-based SD-TEVGs with myofibroblasts [63]. After implantation as femoral artery bypass grafts for 90 to 195 days, the patency was 83%, 75% and 0% for bare polyethylene, wrapped in PGA mesh or polypropylene mesh, respectively. The grafts failed due to thrombus regardless the material of synthetic tubing mold inserted and the use of antithrombotic treatment. Similarly, Rothuizen et al. inserted a rod composed of copolymers (PEOT/PBT) that was wrapped by an external PCL sheet subcutaneously for 4 weeks [71]. The obtained autologous IBTA-based SD-TEVGs manifested 88% patency after 28 days as carotid artery grafts, and the failure of the grafts was attributed to peri-anastomotic intimal hyperplasia, which is different from that reported by Chue and coworkers [63]. Moreover, Wang et al. inserted the PTFE rod subcutaneously and generated autologous IBTA-based SD-TEVGs, which were further decellularized and coated with heparin [73]. These decellularized autologous IBTA-based SD-TEVGs, used as carotid artery bypass grafts, were reported to have a comparable patency (67% after 60 days) to the abovementioned non-decellularized IBTA-based SD-TEVGs [63,71]. Still, grafts failed due to anastomotic stenosis and resulting thrombus, but exhibited excellent mechanical strength in vivo. Mechanical stability is also observed by Furukoshi et al., that the subcutaneously generated autologous IBTA-based SD-TEVGs manifested 100% patency and no aneurysm formation or hemorrhage as arteriovenous shunt within the 28 days follow-up even with repeat percutaneous puncture [83]. Besides the autologous IBTA-based SD-TEVGs, Nakayama et al. developed allogeneic IBTA-SD-TEVGs by inserting nylon mold subcutaneously and decellularized with 70% ethanol [52]. Similar to the autologous IBTA grafts, the allogeneic IBTA-based SD-TEVGs fixed by ethanol showed favorable patency (100% patency after 30 days), both as carotid artery bypass grafts and arteriovenous shunts with application of antithrombotic treatment.

Thus overall, as arterial grafts, IBTA-based SD-TEVGs [52,63,71,73], which are from allogeneic [52] and autologous [63,71,73] origins, display relative higher patency than in-vitro-developed hybrid SD-TEVGs [64,77] that contain xenogeneic natural polymers, when both used with antithrombotic treatment and no endothelial recellularization. This suggests that the nonxenogeneic origin of ECM is important to improve the in vivo patency of SD-TEVGs. Nevertheless, it may also indicate that myofibroblasts transdifferentiated into SMCs have an impact [63,124]. Since current IBTA-based SD-TEVGs lack EC seeding [52,63,71,73], it would be of great interests to investigate if EC seeding may improve in vivo patency of IBTA-based SD-TEVGs.

### 3.3. Modification on SD-TEVGs

As we have described above, many SD-TEVGs still fail, but chemical, biological and mechanical modifications may improve patency in the future. Indeed, a recent meta-analysis emphasized that such scaffold modifications improve patency, at least in large animals [122].

In surface modification, agents or bioactive molecules are attached or conjugated onto scaffolds by either physical or chemical treatments. This process reduces thrombogenicity and increases blood compatibility conferring general antithrombotic characteristics to the SD-TEVGs [131]. The most popular molecules used for surface modification embrace those possessing antithrombotic properties such as heparin, nitric oxide, tissue-type plasminogen activator (t-PA), thrombomodulin, prostacyclin, and their analogues [132]. In the retrieved large-animal studies, four studies exploited heparin coating of the TEVGs (Table 4, Table 5, Table 6 and Table 7). As such, PCL graft was coated with heparin and tested without systemic antithrombotic treatment [60]. The coated PCL graft showed 100% patency after 28 days, which is similar to that observed in pure PCL tested with systemic antithrombotic treatment [45], suggesting that antithrombotic coating is effective. However, the former study was performed in a less challenging anastomosis fashion (ETE). Additionally, heparin has been coated onto decellularized autologous IBTA [73] and decellularized allogeneic canine carotid arteries combined with VEGF coating [75] and EC seeding [76], all showing favorable patencies (67–95% after 60–180 days). However, since the combinations were tested in vivo under systemic antithrombotic treatment, the effectiveness of heparin coating alone is difficult to ascertain. Moreover, surface modifications tested that promote the adhesion and retention of ECs include coating with type I bovine collagen [70], fibronectin, alpha-2(VIII) collagen [72], POG7G3REDV peptide [49], P15 cell-binding peptide [47], and antihuman CD34 monoclonal antibodies [80]. Specific coating-like factor XIII has been explored to increase the crosslinking and strength of highly compacted fibrin TEVG [54], and semisynthetic heparan sulphate-like coating showed antiadhesive properties and seems to prevent neointimal formation [74]. Nevertheless, new approaches are continuously developed and include other molecules as well.

Biological modification refers to recellularization by in vitro repopulation of a scaffold with ECs and/or SMCs. This biomimics the native blood vessel structure and is therefore expected to improve SD-TEVG patency. It is well known that ECs contribute to thrombosis, as well as anticoagulant and fibrinolysis events under physiological conditions. Already, endothelialization has been shown to reduce SD-TEVG thrombogenicity and to improve outcomes [122,133]. Whereas ECs seem obligatory for in vivo SD-TEVG function, SMCs might be dispensable [122]. Among the studies included (Table 4, Table 5, Table 6 and Table 7), and described above, ECs were usually isolated from veins [46,56,58], derived from fat [62], or differentiated from progenitor cells or stem cells from peripheral blood [54,57,78], or bone marrow [55,61]. SMC recellularization has only been applied in a few studies [46,54,55,56,57,61,64,69], in which autologous SMCs were isolated from artery or vein or differentiated from progenitor cells in peripheral blood or bone marrow and seeded on the scaffold. However, the effectiveness of SMC recellularization alone is hard to define, since all these studies combined SMCs with ECs for recellularization. However, since arterial grafts in CABG patients has been reported to have a higher patency rate than venous grafts, SMCs might strengthen grafts and improve long-term patency [134]. The vigorous contribution from SMCs in modulating the graft is evidenced in a rat carotid model [135], where SMC seeding affected extracellular matrix composition in the intima and inhibited intimal hyperplasia [136], the latter being a major factor in medium- and long-term graft failure.

Moreover, to adapt the recellularized endothelial cells for a given arterial pressure and shear stress found in vivo, mechanical modification called preconditioning can be conducted using a peristaltic pump mimicking the palpating blood flow in vivo. Most of the retrieved large-animal studies that displayed excellent patency had seeded scaffolds with ECs [46,51,54,55,56,57,58,61,62,64,65,69,72,76,77,78,81], which might be important for long-term SD-TEVG patency [137]. Alternatively, perfusion bioreactors have been designed to control both media flow and culture environment precisely in order to stimulate the recellularized cells mechanically and chemically [137]. However, most of the preconditioning studies performed in large animals were matching either the shear stress [51,64,65,69,76,77,81] or the pressure [46,78], whereas the graft in the in vivo situation has to withstand both of these two mechanical parameters in parallel to fulfill the physiological requirements. A perfusion bioreactor can imitate flow rate that is related to both shear stress and pressure. However, to achieve the physiological shear stress and pressure level in parallel in the bioreactor, it also requires that the contacting media have a viscosity similar to blood and the distal resistance of TEVG in culture is similar to that of the bypass graft in vivo. The desired viscosity can be achieved using culture medium supplemented with dextran and will result in endothelial properties suited for arteries [138] and the required resistance could be achieved by adding a specific length of tubing after the TEVG in the bioreactor loop system. Moreover, the bioreactors that includes rotational forces enable optimal perfusions throughout the 3-dimensional structure of bioengineered SD-TEVGs [139]. Thus, several modifications may be used to further improve overall patency of the resulting SD-TEVGs.

### 3.4. In Situ SD-TEVGs Recellularization

In vitro recellularization indeed seems to improve patency as reflected in the studies performed in large animals (Table 4, Table 5, Table 6 and Table 7). However, the process is time consuming, which should be taken into consideration. This may also explain why the majority of studies still mainly investigate designs relying on in situ recellularization. There are three in situ recellularization mechanisms involved after implantation of SD-TEVGs: transanastomotic ingrowth, transmural capillarization, and fallout endothelialization [133]. Transanastomotic ingrowth occurs mainly in short TEVGs less than 2 cm with the involvement of EC proliferation from native vessel next to the proximal and distal anastomosis site [140,141,142]. However, this mechanism is insufficient when long grafts are implanted in patients, likely due to limited EC proliferation and senescence [141,142,143]. However, this knowledge is based entirely on synthetic TEVGs, while results await from nonsynthetic TEVGs, such as decellularized TEVGs. Transmural capillarization has been identified as capillaries that sprout from the granulation tissue in adventitia and grow through the graft towards the lumen [144,145]. These capillaries were found to be associated with the generation of endothelium in many synthetic grafts [146,147]. However, it is not easy to present the isolated effect of transmural capillarization, since it is not possible to exclude the role of another endothelialization mechanism referred as fallout endothelialization [148]. In fallout endothelialization, endothelial progenitor cells (EPCs) from the blood stream adhere to the graft through recognizing the protein absorbed on the surface of the graf, and here proliferate and differentiate into mature ECs [149,150]. Transmural capillarization is quite similar to angiogenesis during embryonic development, in which a capillary network is established after endothelial lumenization and fusion or extension of periphery blood islands made by endothelial cells [151]. Fallout endothelialization also seems to mimic developmental angiogenesis, since CD34+ progenitor cells are determined to contribute to both fallout endothelialization and embryological artery formation. During embryogenesis, angioblasts, and hematopoietic cells originate from a common precursor and arise almost at the same time in extraembryonic blood islands. In addition, CD34+ progenitor cells circulating in the blood have been shown to differentiate into endothelial cells [152]. Until now, the understanding of the biological mechanisms in recellularization arising after implantation of SD-TEVGs is still limited. More effort is needed to reveal these mechanisms in healthy as well as in diseased animals and individuals, so that the in vivo outcome of novel SD-TEVGs studies can benefit from the knowledge and further be translated into clinically relevant examinations.

### 3.5. In Vivo SD-TEVG Graft Failure

As described above, none of the current studies has developed an ideal SD-TEVG that do not, at some point, fail through different mechanisms. As a golden standard graft for CABG, saphenous vein is used but fails due to thrombosis in the early phase, whereas intimal hyperplasia and atherosclerosis underline the failure in the intermediate and late phases, respectively [7]. SD-TEVG failure mechanisms during in vivo large-animal studies more or less equal to those found for saphenous vein [153,154]. Thrombus formation is primarily associated with low blood compatibility, compliance mismatch, exposure of a prothrombotic composition, lack of functional ECs, and poverty of proper antithrombotic molecules [41,60,155,156]. Intimal hyperplasia, on the other hand, causes SD-TEVG failure usually at medium term after implantation and reflects a range of factors (damaged ECs, inflammatory response against the graft material, deposition of micro thrombus, and change of hemodynamic forces). Such events might trigger the release of growth factors and further damage the ECM, which stimulates SMC proliferation and migration towards the intima layer and might also lead to deposition of new ECM, which finally narrows the SD-TEVG lumen [153,157]. SMCs involved in intimal hyperplasia are suggested to originate from (1) SMCs that switch from a “contractile” to a “synthetic” SMCs, or (2) circulating precursors (e.g., mesenchymal stem cells, bone marrow derived progenitor cells, and monocytes), or (3) adventitia derived fibroblasts [157,158,159]. Through interaction with glycosaminoglycan in the ECM, intimal hyperplasia further accelerates lipoprotein retention, which develops into atherosclerotic plaques [160,161]. Being highly susceptible to thrombus, atherosclerotic plaques thus increase the risk of graft occlusion and is a major reason for SD-TEVG failure. In contrast, graft rupture or aneurysm is less frequent, since they can be avoided by mechanical testing before in vivo evaluation in animal models. However, mechanical strength can still be weakened after implantation due to rapid scaffold degradation or insufficient ECM deposition by host cells. In the large-animal studies, mechanical failures (Table 4, Table 5, Table 6 and Table 7) such as rupture [54,79], prosthetic kink [68,81], delamination [74], or dilation [57] have been reported, however they are not observed in the rest of the studies. Thrombus remains the dominant failure mechanism observed until now and may be reduced through application of systemic antithrombotic treatment [44,45]. The second often seen failure mechanism is intimal hyperplasia (Table 4, Table 5, Table 6 and Table 7) [50,71,73,74], occurring at the anastomosis site. Regarding allogeneic native ECM-based SD-TEVGs, some studies report thrombus as failure [55,61], whereas our sheep study reports that the thrombus was absent but instead failure was due to distal anastomotic stenosis resulting from intimal hyperplasia in the allogeneic dSCA [50]. Such stenosis however leads to thrombus in the end because of blood stasis according to “Virchow’s triad”. It would be desirable if the existences of intimal hyperplasia could be clearly distinguished from thrombus occurring in the intermediate phase [75,76]. In contrast to thrombus and intimal hyperplasia, atherosclerosis has not been reported as a reason for graft failure in large animals yet (Table 4, Table 5, Table 6 and Table 7). This is most likely explained by the fact that atherosclerosis only becomes relevant upon long-term patency, which has not yet been achieved for SD-TEVGs in large animals.

Thus, in general only a few causes underlie SD-TEVG graft failure, which may seem simple in nature of prevention. However, in reality these challenges have persisted after decades of research and still needs to be solved to achieve clinical success. Whereas proper mechanical strength seems implemented with mechanical failures like ruptures being seldomly reported in large-animal studies, further improvement of biocompatibility to prevent thrombus and intimal hyperplasia is required.

## 4. Conclusions and Perspectives

Until now, a large range of materials have been tested as SD-TEVGs, including synthetic polymers, natural molecular, or a combination of these, as xenogeneic or allogeneic grafts, with or without recellularization of autologous cells. However, studies with high clinical relevance, such as those performed in large-animal models, are still limited in numbers, and most of the studies provide little success: either by limited patency of the grafts or very short observation time. Long-term (>12 months) patency of the grafts is generally not observed except for one study reporting PEtU–PDMS-based SD-TEVGs manifesting 100% patency even after 2 years with application of both end-to-side anastomosis and antithrombotic treatment [53]. Despite being published already back in 2010, no further clinical study has been reported for these PEtU–PDMS-based SD-TEVGs.

Although, we have tried to categorize large-animal studies testing SD-TEVGs for comparison, direct comparisons are indeed difficult, since SD-TEVG and study designs vary a lot. Two parameters important for clinical translation relate to the anastomotic fashion (ETS and ETE) and the systemic antithrombotic treatment. ETS anastomosis is used in CABG and other bypass surgeries, while ETE anastomosis is much less used in clinical settings. As compared to ETE anastomosis, ETS anastomosis represents a more challenging situation, since complicated hemodynamic characteristics provoke adverse biological responses from the surrounding tissue and blood components. This in turn might influence intimal hyperplasia and thrombus formation, the two most common graft failure mechanisms. For instance, Anderson et al., have reported 66% patency when using ETS anastomosis as compared to 100% patency for ETE anastomosis [162]. Considering that 25 out of 32 large-animal studies performed ETE surgery when testing SD-TEVGs as arterial bypass grafts, it seems reasonable to speculate that reported SD-TEVG patency in general may be exaggerated in the field, and that ETS should be considered for future studies.

Likewise, and as mentioned several times above, systemic application of antithrombotic therapy is a commonly used treatment after bypass surgery in clinical practice due to its substantial improvement on patency [163]. This is also clearly supported in the identified literature of large-animal studies testing SD-TEVGs, and 66% (26 out of 39) of large-animal studies have already adopted antithrombotic treatment (Table 4 and Table 6). In particular, direct comparisons have shown that pure PCL SD-TEVGs exhibit a 100 versus 0% patency upon antithrombotic treatment after 4 weeks [45]. Likewise, Weber et al. reported a 67% patency of a bacterial cellulose tube on day 270 with systemic antithrombotic treatment, while bacterial cellulose tube without systemic antithrombotic treatment displayed failure already 56 days after implantation [44]. Thus, even though the lack of antithrombotic treatment may seem ideal, the field may overlook important candidate SD-TEVGs, if systemic antithrombotic treatment is avoided. Moreover, it should be noted that occlusions still occur under systemic antithrombotic treatment, especially when the observation period is extended [62,68,74]. Long-term follow up seems therefore required to reveal the effect of experimental factors on graft patency when systemic antithrombotic treatment is involved.

Besides the testing model-related parameters, graft material and modifications are determinants for graft outcome in large-animal models. Graft rupture due to weak strength is still an observed failure mechanism, especially in natural polymer- and cell-secreted ECM-based SD-TEVGs, whereas synthetic SD-TEVGs can be fabricated with adjustable compliance and strength. However, natural SD-TEVGs, especially nonxenogeneic ones, display higher biocompatibility than synthetic SD-TEVGs, and they are therefore more protected from thrombus, which is the dominant type of graft failure in large-animal models until now. In native ECM-based SD-TEVGs and hybrid SD-TEVGs, xenogeneic originated molecular or matrix seems to harm the graft outcome. Thus, in vitro developed hybrid SD-TEVGs using allogeneic banked human cells or isolated autologous stem cells would be an optimal choice to avoid xenogeneic material. Beside this, it seems that allogeneic native ECM-based SD-TEVGs and IBTA-based SD-TEVGs of autologous or allogeneic origins may be adequate choices as well. However, allogeneic human grafts should be obtained from either cadaveric donor or from waste sources such as dHUA SD-TEVGs. Tissue of origin and unknown underlying diseases may affect SD-TEVG quality, and be uncontrollable factors increasing the cost to SD-TEVG fabrication in a large scale. Besides the choice of graft origin, modifications such as autologous ECs seeding, and preconditioning are showing significant effect in maintaining graft patency and are therefore strongly recommended prior to implantation. To further manipulate the SD-TEVGs and ameliorate the in vivo outcome, solutions to issues such as stem cell sources for repopulating the media compartment and substituting the ECs would also be of great value. Moreover, the association between the ECM ultrastructure and vessel cell retention is important as are signaling between host cells and graft components and the continued remodeling of grafts after implantation.

Thus, the field of SD-TEVG is continuously evolving, but even after decades of research, several challenges remain and require further investigation before the concept of SD-TEVG can be translated into standard care in humans.

## Figures and Tables

**Figure 1 cells-10-00713-f001:**
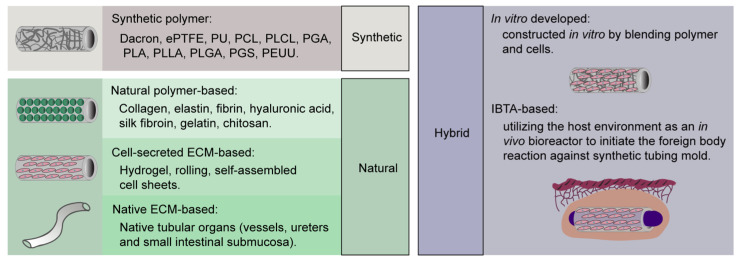
Classification of SD-TEVGs tested in large animals as arterial bypass graft or arteriovenous shunt with length ≥ 10 times the diameter. IBTA: in-body tissue architecture.

**Table 3 cells-10-00713-t003:** Literature review strategies for SD-TEVGs in large animals.

	Inclusion Criteria	Exclusion Criteria
1	In vivo in large animal (>rabbit)	In vitro or in small animal (≤rabbit) or in human
2	Inner diameter ≤ 6 mm	Inner diameter > 6 mm
3	Bypass at small-/medium-diameter artery site (E.g.: coronary, carotid or femoral artery)	Bypass at large-diameter artery site or venous system (E.g.: aorta, aortoiliac artery bypass pulmonary vein or artery, cavopulmonary connection, and venous bypass)
4	Graft evaluated as arterial bypass graft or arteriovenous shunt	Graft evaluated as microvascular network, microvessels, stent, valve, or patch
5	Graft length ≥ 10 times of diameter	Graft length < 10 times of diameter

**Table 9 cells-10-00713-t009:** Advantages and disadvantages of SD-TEVGs in different material.

SD-TEVG Type	Advantages	Disadvantages
**Synthetic SD-TEVGs**	Easy availability and customization.	Biological incompatibility and thrombogenicity.
**Natural SD-TEVGs**	Low immunogenicity and high biological compatibility (allogeneic). Adequate mechanical strength (native ECM-based).	Immunogenicity and thrombogenicity (xenogeneic) Weakness in mechanical strength (natural polymer-based). Limited availability (native ECM-based).
**Hybrid SD-TEVGs**	Biological compatibility Adequate mechanical strength Availability	Thrombogenicity when incorporated with xenogeneic natural polymers. Short off-shelf time and unavailability for acute procedures.

## Data Availability

No new data were created or analyzed in this study. Data sharing is not applicable to this article.

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
