# Peer review of "Review: Tissue Engineering of Small-Diameter Vascular Grafts and Their In Vivo Evaluation in Large Animals and Humans"

_cells, 2021, doi:10.3390/cells10030713_

Round 1

Reviewer 1 Report

The review article by Fang et al, is a comprehensive review on the evolution of the field of vascular grafts engineering and their in vivo evaluation. I enjoyed a lot reading this review article as it covers the subject adequately and has comprehensive tables that summarise all the strategies described which is very useful for the reader. Few suggestions

1) I found that a general figure/schematic to show the scope of the review, is missing. (i.e., check two indicative similar reviews doi: 10.3390/app9071274, doi: 10.3389/fbioe.2018.00041)

2) Similarly, figures from some of the key papers presented are missing.

Without any figures the review feels "thin".

3) The role of growth factors and other "signals" and/or environmental cues could be improved as I feel it is not adequately mentioned  

I generally recommend its publication. 

Author Response

Response to Reviewer 1 Comments

Point 1: I found that a general figure/schematic to show the scope of the review, is missing. (i.e., check two indicative similar reviews doi: 10.3390/app9071274, doi: 10.3389/fbioe.2018.00041)

Response 1: We thank the reviewer for this suggestion, and the schematic figure (Fig. 1) has been added accordingly to show the scope of the review.

Point 2: Similarly, figures from some of the key papers presented are missing. Without any figures the review feels "thin".

Response 2: We acknowledge the reviewer for his/her point. However, since we attempted to perform an unbiased review the studies included were screened by the inclusion and exclusion criteria systematically, and they are discussed and compared to draw the conclusion. Therefore, they are considered equally important and it would be inappropriate to define specific key papers as such. We hope the reviewer can acknowledge that.

Point 3: The role of growth factors and other "signals" and/or environmental cues could be improved as I feel it is not adequately mentioned.  

Response 3: We agree and have therefore inserted the role of growth factors and other chemical modifications involved in small diameter TEVGs tested in large-animals in section “3.3 Modification of SD-TEVGs”, and hope the reviewer now finds this sufficiently described. We hope the reviewer finds this section improved.

Reviewer 2 Report

Overall, this was an extremely well written, informative, and comprehensive review paper that spanned large animals to humans. The only shortcoming was not including any information regarding vascular graft studies in rodents, which are widely studied. It is understandable that this was not done, as the focus of the review paper was in large animals to humans. As a service to the reader it would have been useful to include rodent studies as a citation (no data or in depth analysis needed) as an alternate to the described work in large animals and humans, when appropriate. That is the only flaw in the paper, which I consider to be a minor flaw.

Author Response

Response to Reviewer 2 Comments

 Point 1: The only shortcoming was not including any information regarding vascular graft studies in rodents, which are widely studied. It is understandable that this was not done, as the focus of the review paper was in large animals to humans. As a service to the reader it would have been useful to include rodent studies as a citation (no data or in depth analysis needed) as an alternate to the described work in large animals and humans, when appropriate. That is the only flaw in the paper, which I consider to be a minor flaw.

Response 1: We thank the reviewer for this suggestion and have now added some citations of papers for small animals like rodent and rabbit studies accordingly.

Reviewer 3 Report

Comments to the Authors (Cells-1125960)

The review would be nice to see other/current update sets for tissue engineered vascular grafts, but there are some concerns. After address these points, I hope that the revised manuscript will be acceptable for publication in Cells.

Suggestions for improvement:

1) Schematic diagramillustrating various types of tissue engineered vascular grafts (TEVGs)improves the understanding of TEVGs.

2) It is helpful to show a “Table” that summarizes the advantages and disadvantagesof each TEVG.

Minor point:

Font size in Table 4 was not properly.

Author Response

Response to Reviewer 3 Comments

 Point 1: Schematic diagram illustrating various types of tissue engineered vascular grafts (TEVGs) improves the understanding of TEVGs.

Response 1: We thank the reviewer for this suggestion, and a schematic figure (Fig. 1) has been added accordingly to illustrate various types of tissue engineered vascular grafts.

Point 2: It is helpful to show a “Table” that summarizes the advantages and disadvantages of each TEVG.

Response 2: We agree with the reviewer, and the Table 6 has now been added accordingly to the manuscript.

Point 3: Font size in Table 4 was not properly.

Response 3: We thank the reviewer for pointing that out and we have now changed the font size to 8 in Table 4.

Reviewer 4 Report

Congratulation on a well written review

Author Response

Response to Reviewer 4 Comments

 Point 1: Congratulation on a well written review.

Response 1: We thank the reviewer for his positive comment.